# Structure and thermodynamics of water adsorption in NU-1500-Cr

Ching-Hwa Ho [1✉], Mason L. Valentine[1], Zhijie Chen[2], Haomiao Xie[2], Omar Farha[2], Wei Xiong [1,3✉] & Francesco Paesani [1,3,4✉]

Metal-organic frameworks (MOFs) are a class of materials with diverse chemical and structural properties, and have been shown to effectively adsorb various types of guest molecules. The mechanism of water adsorption in NU-1500-Cr, a high-performance atmospheric water harvesting MOF, is investigated using a combination of molecular dynamics simulations and infrared spectroscopy. Calculations of thermodynamic and dynamical properties of water as a function of relative humidity allow for following the adsorption process from the initial hydration stage to complete filling of the MOF pores. Initial hydration begins at the water molecules that saturate the open $Cr^{3+}$ sites of the framework, which is then followed by the formation of water chains that extend along the channels connecting the hexagonal pores of the framework. Water present in these channels gradually coalesces and fills the hexagonal pores sequentially after the channels are completely hydrated. The development of hydrogen-bond networks inside the MOF pores as a function of relative humidity is characterized at the molecular level using experimental and computational infrared spectroscopy. A detailed analysis of the OH-stretch vibrational band indicates that the low-frequency tail stems from strongly polarized hydrogen-bonded water molecules, suggesting the presence of some structural disorder in the experimental samples. Strategies for designing efficient water harvesting MOFs are also proposed based on the mechanism of water adsorption in NU-1500-Cr.

[1] Department of Chemistry and Biochemistry, University of California San Diego, La Jolla, CA 92093, USA. [2] Department of Chemistry and International Institute of Nanotechnology, Northwestern University, 2145 Sheridan Road, Evanston, IL 60208, USA. [3] Materials Science and Engineering, University of California San Diego, La Jolla, CA 92093, USA. [4] San Diego Supercomputer Center, University of California San Diego, La Jolla, CA 92093, USA. ✉email: c9ho@ucsd.edu; w2xiong@ucsd.edu; fpaesani@ucsd.edu

With a changing climate, water scarcity has become one of the most pressing global issues, which already affects almost two thirds of the world's population[1]. Various technologies (e.g., membrane and thermal desalination) have been developed to produce freshwater from seawater[2,3]. Due to the associated cost and infrastructure required, most of these technologies, however, are generally only viable for large-scale and centralized water production. Decentralized water production represents an alternative strategy to provide freshwater to areas that cannot be easily connected to a centralized distribution network. Since air contains $\sim 10^{21}$ liters of water as drops and vapor, atmospheric water harvesting (AWH) has emerged as a promising alternative approach to provide freshwater to areas of water scarcity, as well as for the development of other water-based technologies[4].

Among porous materials that can efficiently adsorb water from air over a tunable range of relative humidity (RH), metal-organic frameworks (MOFs) have recently attracted particular interest[5–7]. Built from organic linkers and secondary building units (SBUs) composed of metal ions or clusters, MOFs exhibit large surface areas and a variety of physicochemical properties that can be tuned for specific applications through either pre- or post-synthetic approaches[8,9]. To act as an efficient water sorbent, a given MOF must satisfy the following requirements: (i) high hydrolytic stability for recycling performance, (ii) large porosity and surface area for high water vapor uptake, (iii) relatively mild regeneration conditions, (iv) adsorption isotherm with a steep uptake at a specific RH value, and (v) high deliverable capacity. Several MOFs with high stability and water sorption capacity have been reported in the literature[10–18]. While measurements and calculations of isotherms and enthalpies of adsorption provide information about the overall performance of a given MOF for water-sorption applications, a molecular-level understanding of the sorption mechanisms, which is key to the design of new MOFs with improved sorption capacities, remains elusive[5–7]. The major difficulty arises from the complexity of the water-framework interface which makes the realistic modeling of the adsorption mechanism particularly challenging. Several models have been developed to investigate the properties of water using computer simulations, which often rely on pairwise additive energy expressions that are empirically parameterized to reproduce a subset of experimental data (e.g., the TIPnP[19–21], SPC*[22,23] families of water models). Due to the difficulties in getting "the right results for the right reasons" in computer modeling of water as a function of temperature and pressure, establishing a reliable connection between computer simulations and experimental measurements of water confined in MOFs remains challenging. Another challenge arises from the intrinsic complexity of the MOF structures. Several water-adsorbing MOFs possess frameworks with various non-equivalent pores, which makes it difficult to unambiguously characterize the water sorption mechanisms at the molecular-level[24–26]. Additional difficulties in computer simulations of water adsorption in MOFs are related to the approximations that are made for modeling the frameworks. Since MOF structures are determined by crystallography, molecular simulations typically only model perfect frameworks in which the unit cell is exactly repeated in each direction without including any defects. Although great efforts have been made to characterize defects in MOFs, little direct information is available about the chemical structure around defects[27–32]. In the context of computer simulations, ambiguities in the framework structure may lead to inaccurate representations of water-framework interactions which, in turn, may affect the overall description of the sorption mechanisms. Other experimental techniques that are sensitive to defects could greatly improve our understanding to defects in MOFs and their interactions with water molecules.

In this study, we investigate the water sorption mechanism of NU-1500-Cr[18], a high-performance MOF for AWH applications, using a combination of computational and spectroscopic approaches. NU-1500-Cr contains hexagonal pores connected by two types of channels that extend along orthogonal directions. Its well-defined single-step adsorption isotherm exhibits maximum capacity of water uptake exceeding 1.0 g/g, which remains effectively unchanged after 20 adsorption-desorption cycles[18]. The molecular mechanisms of water adsorption in NU-1500-Cr are unravelled from systematic analyses of both thermodynamic and dynamical properties of water that are calculated from advanced molecular dynamics (MD) simulations carried out with the MB-pol many-body potential[33–35]. We determine that the adsorption process begins with water molecules saturating the Cr(III) open sites of the framework and proceeds with the formation of water chains along one of the channels, which is then followed by sequential filling of the main pores.

## Results and discussion

**Thermodynamics of water adsorption.** The experimental adsorption isotherm of water in NU-1500-Cr (Fig. 1A) slowly increases up to ~33% RH where it exhibits a sharp step that effectively ends at ~37% RH, after which it only shows a moderate increase up to ~70% RH. It should be noted that the adsorption isotherm shown in Fig. 1A slightly differs from the adsorption isotherm reported in ref. [18]. The difference is due to the use of an incorrect value of the saturated vapor pressure of water at 298 K in the original study, which results to the original adsorption isotherm being shifted to slightly higher RH values relative the adsorption isotherm measured in the present study. To gain insights into the confining effects of the framework on the properties of water adsorbed in the pores, the enthalpy of adsorption ($\Delta H_{ads}$) and water

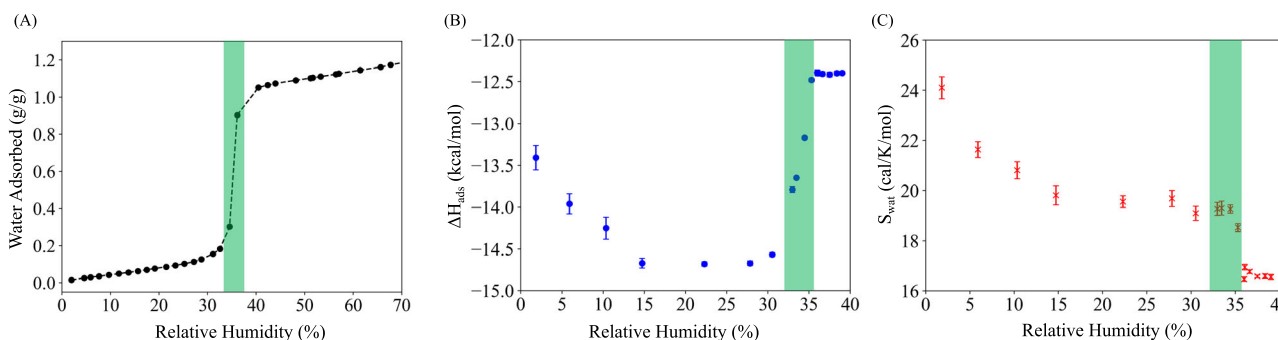

**Fig. 1 Thermodynamics of water adsorption in NU-1500-Cr. A** Experimental water adsorption isotherm. **B** Enthalpy of adsorption $\Delta H_{ads}$ and (**C**) water entropy $S_{wat}$ at different RHs. Green regions correspond to the adsorption step.

entropy ($S_{wat}$) calculated from MD simulations carried out as a function of RH are shown in Fig. 1B, C respectively. $\Delta H_{ads}$ becomes progressively more negative, indicating stronger framework–water interactions, as the RH increases and plateaus at a value of ~ − 14.5 kcal/mol between 15% and 31% RH. After this point, $\Delta H_{ads}$ rapidly increases to a value of ~ − 12.5 kcal/mol in correspondence of the step in the adsorption isotherm between ~33% and ~37% RH and remains approximately constant at higher RH. At the lowest RH, the calculated enthalpy of adsorption is −13.4 kcal/mol, which is in a good agreement with the experimental values extracted from measurements of water adsorption in other MOFs using the Clausius-Clapeyron equation[10,13,36,37]. The decrease in $\Delta H_{ads}$ observed from 1.8% to 31% RH can be attributed to both the development of hydrogen bonds among water molecules and framework–water interactions. Since the hydrogen-bond network becomes more extended as the RH increases, the increase in enthalpy in correspondence of the adsorption step (i.e.~33% and ~37%) suggests that framework–water interactions play a minor role at high RH values. It is worth noting that $\Delta H_{ads}$ at high RH values is more negative than the enthalpy calculated from MB-pol simulations of liquid water at 298 K ( ~ − 10.96 kcal/mol), which suggests that the confinement of water molecules within the NU-1500-Cr framework is, overall, energetically favorable.

$S_{wat}$ displays similar variation to $\Delta H_{ads}$ up to ~31% RH, decreasing monotonically to a value of ~19.6 cal/mol K. Contrary to $\Delta H_{ads}$, $S_{wat}$, however, exhibits a steep drop at ~36% RH, and remains approximately constant to a value of 16.5 cal/mol K at higher RH. The decrease in entropy as a function of RH is the result of both confinement effects caused by the framework and spatial constraints due to hydrogen bonding which restrict the motion of water molecules. Importantly, $S_{wat}$ approaches a value of 16.5 cal/mol K at high RH beyond the adsorption step, which is very close to the entropy of liquid water at the same thermodynamic conditions ($T = 298.15$ K and $P = 1$ atm) 16.7 cal/mol K[38]. Although this similarity may suggest liquid-like behavior for water in the NU-1500-Cr pores at high RH, systematic analyses of local structure and hydrogen-bonding topologies reported in the next sections indicate that there are significant differences. The shape of the adsorption isotherm can directly be rationalized in terms of the competition between enthalpic and entropic effects of water. At low RH, $\Delta H_{ads}$ compensates for the loss of entropy that accompanies the transition of water molecules from the gas phase to the NU-1500-Cr pores and then becomes the dominant factor during pore filling that is triggered by the formation of an extended hydrogen-bond network among the water molecules. A similar trend in enthalpy of adsorption as a function of RH was observed in UiO-66 and its functionalized derivatives, which have been well-known as atmospheric water harvesters[39].

**Structure and filling mechanism**. To characterize the thermodynamic properties of water in NU-1500-Cr at the molecular level, we examine the structure of the framework and the distribution of water at different RH values. Figure 2(A, B) illustrates the structure of the empty framework, which is composed of $Cr_3(\mu\text{-}O)(H_2O)_2Cl$ SBUs linked by deprotonated peripherally extended triptycenes (PETs). Two $Cr^{3+}$ metal ions in each SBU are saturated with water molecules, while the third $Cr^{3+}$ metal ion is saturated with a chloride ion. Each SBU is connected with three PETs that build up a three-dimensional network with the 1-dimensional hexagonal pores of diameter ~1.4 nm extending along the $c$-axis, as shown in Fig. 2(A). The other two types of channels (hereafter referred to as type-A channel and type-B channel) orthogonal to the hexagonal pores allow water molecules to move across the hexagonal pores. A type-A channel is

composed of two adjacent SBUs, which are parallel to the hexagonal pores, and the aryl groups of three PETs. A type-B channel is created from the stacking of two PETs along the direction of the hexagonal pores (Fig. 2(B)).

Figure 3(A) summarizes the statistics of different hydrogen-bonding topologies created by water molecules as a function of RH. Each water molecule is classified as donor ($n$D) or acceptor ($m$A) according to the type and number ($n$ and $m$) of hydrogen bonds it engages in. The majority of water molecules acts as no donors-single acceptors (i.e., 0D-1A) at the lowest RH (1.8%), which suggests there exist interaction sites in the framework that serve as hydrogen-bond donors. Analyses of oxygen-oxygen radial distribution functions (RDFs) presented in Supplementary Fig. 5 demonstrate that the first water molecule that enters the framework directly binds to one of the water molecules that saturate a $Cr^{3+}$ open site. Therefore, the saturating water molecules are the primary interaction sites for initial hydration of the MOF pores. Similar conclusions were drawn from computer simulations of water in MIL-100(Fe), MIL-101(Cr), and $Co_2Cl_2BTDD$, and a recent experimental study of water in HKUST-1[40–43]. This similarity in the initial stages of the hydration process is associated with similarly strong interactions between water and the open metal sites of the frameworks, which give rise to dative bonds between the oxygen lone-pair of a water molecule and an empty d-orbital of the metal ions. It should be noted that the presence of open metal sites in the framework is not a necessary condition for a MOF to be able to adsorb water, since functional groups can also serve as interaction sites. For example, MOF-801 and ZIF-90 adsorb water via their $\mu_3$-OH and carbonyl groups, respectively[10,44].

Figure 3(A) indicates that the 1D-1A hydrogen-bonding topology becomes dominant as the RH increases beyond 15%, while the 2D-2A topology is dominant after the adsorption step (i.e., at RH higher than 37%). As water molecules get adsorbed, they primarily form water chains that emanate from the saturating water molecules and extend into type-A channels. The spatial distribution of the water chains inside the framework at 14.7% RH shown in Fig. 2(C, D) indicates that the water molecules are located in type-A channels and connect the adjacent SBUs, while leaving both the hexagonal pores and type-B channels empty. Similar spatial distributions are obtained for all RH values below 33%, indicating that type-A channels have sufficient space to accommodate up to 0.24 g/g of water. It should be noted that the relatively small diameters (~9.6 Å) of type-A channels prevent the water molecules from forming extended hydrogen-bond networks as in liquid water, which implies that favorable water–framework interactions must be responsible for the adsorption of water inside type-A channels at the early stages of the adsorption process. These results can be rationalized by considering the water-benzene dimer as a model system for describing the interactions between water and the PET linkers of type-A channels. As shown in Supplementary Fig. 4, the benzene-water potential energy scan exhibits a binding energy of ~2 kcal/mol when the water molecule point one of the OH bond towards the aromatic ring, which is consistent with previous ab initio calculations of benzene-water interactions[45]. As a consequence, the more negative values calculated for the enthalpy of adsorption from 1.8% to 31% RH can be attributed to the growing number of water molecules interacting with the PET linkers of type-A channels.

At RH values higher than 33%, the additional water molecules start filling the hexagonal pores as type-A channels are fully occupied, which corresponds to the adsorption step on the isotherm, and the abrupt changes in adsorption enthalpy and water entropy shown in Fig. 1(B), (C). Pore filling begins with the formation of water bridges connecting water molecules located in

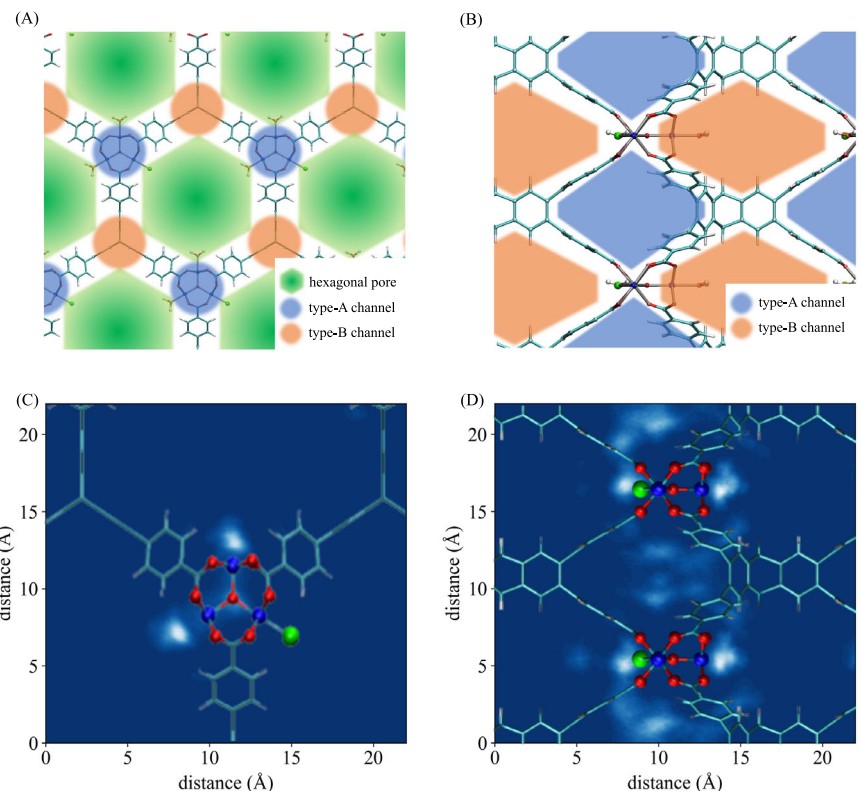

**Fig. 2 NU-1500-Cr structure and water density distribution. A** Structure of NU-1500-Cr viewed along the hexagonal pore direction. The hexagonal pores are shown in green, while type-A and type-B channels are shown in blue and orange, respectively. **B** Structure of NU-1500-Cr viewed along a direction perpendicular to the hexagonal pore direction. The hexagonal pores are not displayed for the clarity, while type-A and type-B channels are shown in blue and orange, respectively, as in (**A**). Two-dimensional water density distribution calculated along (**C**) and perpendicular (**D**) to the hexagonal pore direction at 14.7% RH. Lighter regions correspond to higher water density. Color scheme: C = cyan, H = white, O = red, Cr = blue, Cl = green.

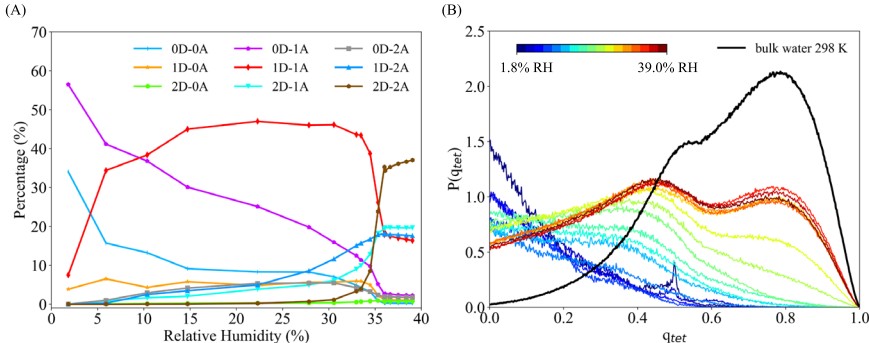

**Fig. 3 Water's hydrogen-bonding structure in NU-1500-Cr. A** Hydrogen-bond topologies distribution. **B** Probability distributions of the tetrahedral order parameter, $P(q_{tet})$. Black curve shows $P(q_{tet})$ of liquid water calculated from MD simulations carried out with MB-pol at 298 K[46].

separate type-A channels, and continues with the development of extended hydrogen-bonding networks that emanate from the water bridges (see Supplementary Fig. 6).

The hydrogen-bonding topologies in Fig. 3(A) also indicate that the dominant topology changes from 1D-1A to 2D-2A as the hexagonal pores start being filled, suggesting that the majority of water molecules experience local environments resembling those found in liquid water. However, the percentage of 2D-2A topology (~37%) is slightly smaller than the corresponding value predicted by MB-pol simulations of liquid water (~45%)[46]. This difference can be attributed to the presence of the framework that constrains the development of hydrogen-bonding networks. Further insights into the structure of water adsorbed in the hexagonal pores of NU-1500-Cr is gained by

analyzing the variation of the tetrahedral order parameter $q_{tet}$ which is defined as[47]

$$q_{tet} = 1 - \frac{3}{8}\sum_{j=1}^{3}\sum_{k=j+1}^{4}\left(\cos\psi_{ijk} + \frac{1}{3}\right)^2, \quad (1)$$

Here, $\psi_{ijk}$ is the angle between the oxygen atom of the central water molecule with index $i$ and two oxygen atoms in the neighbor with index $j$ and $k$ at a distance smaller than 3.5 Å. A value 0 for $q_{tet}$ indicates a completely disordered arrangement of the water molecules (as in an ideal gas), while $q_{tet} = 1$ indicates a perfectly tetrahedral arrangement of water molecules. Figure 3(B) shows that the probability $P(q_{tet})$ of local structures with $q_{tet} > 0.6$ is negligible from 1.8% to 22.3% RH, indicating that a low

relative humidity the water molecules are spatially arranged in a disordered manner. As RH increases, $P(q_{tet})$ acquires a bimodal distribution with two maxima at $q_{tet} \approx 0.4$ and $q_{tet} \approx 0.8$. Compared to $P(q_{tet})$ calculated with MB-pol for liquid water at 298 K (black curve in Fig. 3(B)), $P(q_{tet})$ calculated for water in NU-1500-Cr at high RH displays a wider distribution at low $q_{tet}$ values and significantly lower amplitude at high $q_{tet}$ values, indicating a relatively more disordered hydrogen-bond network compared to liquid water.

From the analysis of the spatial distribution of water in NU-1500-Cr, which is shown in Fig. 4 for 34.6% RH (corresponding to the middle of the step in the adsorption isotherm), it is possible to conclude that the hexagonal pores are filled sequentially. Similar mechanisms were reported for water adsorption in ZIF-90, MIL-100(Fe), and MIL-101(Cr)[40,41,48]. Since the water molecules located in the main pores are far away from the framework and surrounded by other water molecules, the associated interaction energy mostly results from water-water interactions. As a consequence, water molecules fill up one pore at a time, preferentially entering pores that already contain other water molecules which can thus maximize the number of hydrogen bonds.

In summary, the evolution of the adsorption isotherm as a function of RH can be rationalized as a sequence of three stages. The $Cr^{3+}$ metal sites of the SBUs provide the strongest binding sites for water adsorption. Upon saturation of the $Cr^{3+}$ binding sites, given their hydrophilic nature, the metal-bound water molecules become new binding sites for other water molecules at the early stages of the adsorption process. Water chains connecting the metal-bound water molecules then develop prior to the filling of the hexagonal pores. These chains extend along type-A channels where the water molecules can establish favorable interactions with the aromatic rings of the PET linkers. The steep uptake displayed by the adsorption isotherm between 33% and 37% RH result from all type-A channels being fully occupied by water, which thus determines the onset of the filling of the hexagonal pores. The mechanism of water adsorption derived from the MB-pol simulations suggests that in order to move the water adsorption step of NU-1500-Cr to lower RH values, the hexagonal pores should be decorated with hydrophilic functional groups that will prevent water molecules from filling the type-A channels at the early stages of the adsorption process.

**Infrared spectra**. To further characterize the water hydrogen-bond networks in the NU-1500-Cr pores, we examined infrared spectra of water in the MOF pores compared to liquid water, focusing on the bend+libration combination band and OH-stretch vibrational band. The bend+libration combination band is highly sensitive to water-water interactions, shifting to lower frequencies when the hydrogen-bond network of water is disrupted, and shifting to higher frequencies when the hydrogen-bond network of water is highly ordered[49]. As shown in Fig. 5(A), the bend + libration combination band is not well defined at low humidities, but becomes apparent near 30% RH as the number of MOF-bound water molecules increases and water-water interactions become more significant. For all RH values above ~30%, the bend + libration combination band for water in NU-1500-Cr is shifted to much lower frequencies than in liquid water, but the shift is smaller above 40% RH. This indicates a highly disrupted hydrogen-bond network that becomes more similar to liquid water as the pores fill. This qualitative trend matches the trends observed for hydrogen-bonds in the simulations shown in Fig. 3(A) and the trend observed for the tetrahedral order parameters in Fig. 3(B), and suggests that water-water hydrogen-bonds increase throughout the pore-filling process.

The OH-stretch vibrational band of water also reports on the strength of hydrogen bonds, and can be decomposed into multiple peaks to analyze different local environments experienced by water molecules. Figure 5(B) shows the OH-stretch vibrational band of water in NU-1500-Cr measured at humidities between 20% RH and 55% RH compared to the corresponding band measured for liquid water. At low RH, the band consists primarily of an extremely broad tail below 3200 cm$^{-1}$ that is not present in the infrared spectrum of liquid water, and a set of narrow sharp peaks around 3600–3700 cm$^{-1}$, at much higher frequencies than in liquid water. As the humidity increases, the sharp high-frequency peaks and broad low-frequency peaks become less prominent relative to the total OH-stretch band, but are still present at each relative humidity.

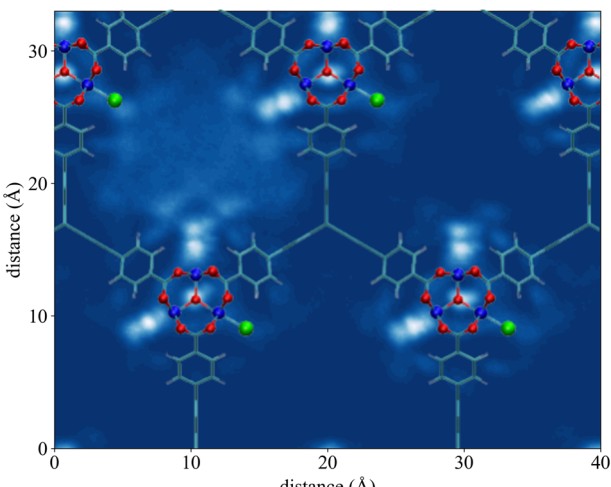

**Fig. 4 Water spatial distribution in NU-1500-Cr.** Two-dimensional water density distribution in the hexagonal pores of NU-1500-Cr calculated at 34.6% RH. Color scheme: C = cyan, H = white, O = red, Cr = blue, Cl = green.

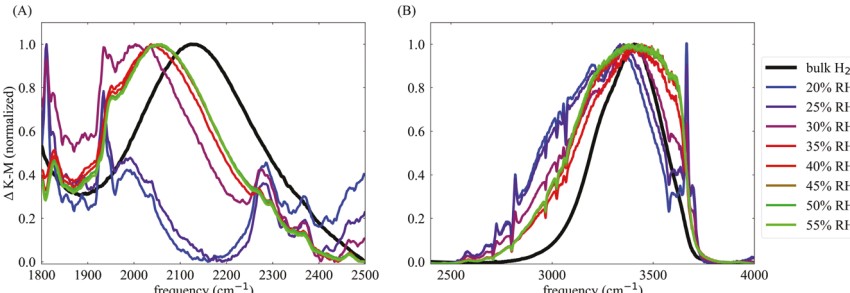

**Fig. 5 Infrared spectra of water confined in NU-1500-Cr as a function of RH compared to bulk water.** Spectra were normalized to the maximum IR signal after subtracting a spectrum of activated NU-1500-Cr to reduce the influence of framework peaks in (**A**) the bend + libration combination band, and (**B**) the OH-stretch region region of the spectrum.

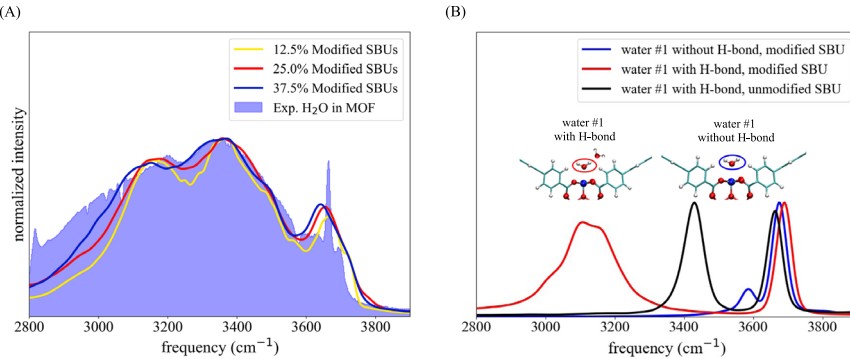

**Fig. 6 Simulated water spectra in NU-1500-Cr with modified SBUs. A** Simulated water spectra with 12.5% (yellow), 25.0% (red), and 37.5% modified SBUs in the framework and experimental water spectrum (shaded purple) at 25% RH. **B** Spectra of water #1 binding to a modified SBU without (blue) and with (red) forming a hydrogen-bond with a neighboring water molecule, and binding to a unmodified SBU with forming a hydrogen-bond with a neighboring water molecule (black).

Control experiments using HOD diluted in $D_2O$ provides simpler lineshapes, which still displays the same three features (see Supplementary Fig. 8). In previous studies of water in confined environments, sharp high-frequency peaks have been assigned to "dangling" and interfacial OH bonds with relatively weak interactions[50–52], while peaks at lower frequencies have been assigned to water molecules in highly charged environments[53,54]. Spectral features between the high-frequency and low-frequency regions were found to more closely resemble those observed for liquid water. Throughout the pore filling process of NU-1500-Cr, the relative intensities of the broad low-frequency features and narrow high-frequency features decreases, and the spectrum progressively becomes more similar to that of liquid water when the pores are completely filled, which is consistent with the calculated hydrogen-bond and $q_{tet}$ results, as well as trend observed for the bend + libration combination band. In the control experiments with HOD (see Supplementary Fig. 8), the trend is even clearer due to the relative simplicity of the OH-stretch vibrational band associated with HOD molecules.

The significant intensity between 2800 and 3100 cm$^{-1}$ at low RH, however, suggests the existence of highly charged atoms polarizing water molecules, which were not originally accounted for in simulations. In order to further elucidate the spectral signatures observed experimentally, Fig. S7 shows the infrared spectrum of water in pristine NU-1500-Cr simulated at 20% RH following the procedure described in ref. [48]. In order to further elucidate the spectral signatures observed experimentally, Fig. S7 shows the infrared spectrum of water in pristine NU-1500-Cr simulated at 25% RH following the procedure described in ref. [48]. While the simulated spectrum overall reproduces the experimental lineshape, the low-frequency tail below 3100 cm$^{-1}$ is missing. This missing tail can be recovered from MD simulations carried out with modified SBUs as shown in Fig. 6(A), where the charge of the $Cr^{3+}$ atoms are increased by 50% and the charges of the oxygen atoms of the carboxylate groups are adjusted accordingly to guarantee charge neutrality. As the fraction of modified SBUs in the MD simulations is increased, the intensity of the low-frequency portion of the simulated spectrum also increases while retaining all the other spectral signatures, which systematically improves the agreement with the experimental lineshape. This analysis suggests that highly charged $Cr^{3+}$ may likely be present in the NU-1500-Cr samples used in the experiments, which may be attributed to undercoordinated $Cr^{3+}$ sites due to structural disorder. The modified SBUs also account for the humidity-dependent trends observed in the relative intensity of the low-frequency features, which are most prominent at low RH when there are relatively few water molecules and they are clustered near the SBUs.

The infrared spectra of the water molecules that directly bind to the $Cr^{3+}$ sites of both unmodified and modified SBUs are calculated in order to isolate their specific signatures from the overall spectra. Figure 6(B) demonstrates that the low-frequency tail in the experimental spectrum stems from the OH-stretch vibrations of the water molecules bound to the charge-modified $Cr^{3+}$ sites (water #1 in the figure) which are involved in a hydrogen-bond with a neighboring water molecule. As shown in Fig. 6(B), while still present in the infrared spectrum of water molecules bound to $Cr^{3+}$ sites with unmodified charges, the spectral feature associated with the hydrogen bond between the metal-bound water molecule and a neighboring water molecule appears at relatively higher frequencies (~3450 cm$^{-1}$).

This analysis demonstrates that the extent of polarization affects the position of the low-frequency features of the infrared spectrum of water in NU-1500-Cr and is responsible for the significant infrared intensity between 2800 and 3100 cm$^{-1}$. This increased intensity can be explained by considering that the MB-pol dipole moment calculated for the water molecules bound to the charge-modified SBUs of NU-1500-Cr is ~4.0 D which is significantly larger than the value of 2.7 D obtained from MB-pol simulations of liquid water[46]. Missing low-frequency tails in the infrared spectra of water adsorbed in $Co_2Cl_2BTDD$ and ZIF-90 can also be observed in previous simulation studies[42,48]. However, in the case of $Co_2Cl_2BTDD$ and ZIF-90 these low-frequency tails are less intense than in the infrared spectra of water in NU-1500-Cr which may possibly be attributed to the lower oxidation states of $Co^{2+}$ and $Zn^{2+}$ compared to $Cr^{3+}$ as well as to different degree of structural disorder in the experimental samples.

## Conclusions

We used advanced molecular dynamics simulations in combination with infrared spectroscopy to investigate the adsorption mechanisms of water in NU-1500-Cr. We found that the confining effects of the framework modulate the thermodynamic properties of water as a function of relative humidity. In particular, the loss of entropy associated with the restricted motion of water molecules inside the NU-1500-Cr pores is compensated by favorable framework-water and water-water interactions at low and high RH, respectively. The variation of both adsorption enthalpy and water entropy directly correlates with the adsorption mechanism. The initial hydration stage occurs with water molecules saturating the $Cr^{3+}$ open sites of the framework. With increasing RH, water molecules first form chain-like structures that emanate from the $Cr^{3+}$-bound water molecules and fill the

narrow channels composed of two adjacent SBUs which are parallel to the hexagonal pores, and the aryl groups of the organic linkers. Due to favorable interactions between water the aryl groups of the framework, type-A channels accommodate all the adsorbed water molecules up to 33% RH. Above 33% RH, filling of the hexagonal pores begins with a steep adsorption step that ends at ~37% RH.

Further insights into the confining effects of the framework and the properties of the hydrogen-bond networks developed by water inside the NU-1500-Cr pores were gained from monitoring the evolution of the infrared spectra of adsorbed water as a function of RH. The bend + libration combination band indicates that the hydrogen-bond networks become more developed as the RH increases, which is consistent with the decrease in water entropy and the formation of more tetrahedral spatial arrangements of water inside the pores. While the simulated spectra reproduce the main features of the experimental lineshapes, they miss intensity in the low-frequency region between 2800 and 3100 cm$^{-1}$. We found that the low-frequency tail in the experimental infrared spectra can be recovered by MD simulations carried out with charge-modified SBUs which can effectively mimic possible structural disorder in the experimental samples. We therefore demonstrated the power of combining MD simulations and IR spectroscopy to gain molecular-level insights into the nature of framework–water interactions which are difficult to obtain from crystallography.

Since the steep uptake in the adsorption isotherm corresponds to the onset of pore filling, our results indicate that decorating the pores with hydrophilic functional groups may prevent water molecules from entering type-A channels at the early stage of the adsorption process which, in turn, can improve the ability of NU-1500-Cr to harvest water from air by shifting the adsorption step to lower RH values.

## Methods

**Material preparation and characterization.** NU-1500-Cr sample was prepared following literature method[18]. The sample was activated at 120 °C for 12 h before sorption mesurements. The N$_2$ adsorption isotherm was measured on a Micromeritics ASAP 2420 (Micromeritics, Norcross, GA) instrument at 77 K (see Supplementary Fig. 10). The pore volume was evaluated at P/P$_0$ = 0.99 as 1.24 cm$^3$/g and the Brunauer-Emmett-Teller (BET) surface area was calculated as 3575 m$^2$/g which correspond well with the reported values (1.24 cm$^3$/g and 3580 m$^2$/g respectively) in the literature. The water adsorption isotherm of NU-1500-Cr was measured on a micromeritics 3Flex at 25 °C (see Fig. 1(A) and Supplementary Fig. 11). Degased Millipore water was used as vapor source. The measurement temperature was controlled with a Micromeritics temperature controller.

**Molecular models.** NU-1500-Cr was modeled using a flexible force field. The structure of NU-1500-Cr was taken from crystallographic data[18], with one Cr$^{3+}$ atom in each SBU coordinated with a chloride ion and the other two Cr$^{3+}$ coordinated with the oxygen atoms of two water molecules. The crystallographic structure was initially optimized in periodic boundary conditions using density functional theory (DFT) calculations carried out with the Vienna Ab initio Simulation Package (VASP)[55–58], using the PBE exchange-correlation functional[59] combined with the D3 dispersion correction[60]. The VASP calculations were carried out using the projector-augmented wave (PAW) method[61,62] with a 700 eV kinetic energy cutoff on a 2 × 2 × 2 k-point grid. The forces were converged to a tolerance of 0.03 eV/Å. The atomic point charges for the force field were obtained using the charge model 5 (CM5)[63] as implemented in Gaussian 16[64] by performing DFT calculations on a cluster model of NU-1500-Cr (see Supplementary Figs. 1-2 for details) using the ωB97X-D functional[65] in combination with the def2-TZVP basis set[66]. The force field parameters for the bonded terms involving the Cr$^{3+}$ atoms were fitted using the genetic algorithm to ωB97X-D/def2-TZVP single point energies calculated with Gaussian 16[64] for 249 distorted configurations of the same cluster model of NU-1500-Cr used in the CM5 calculations. The Lennard-Jones (LJ) coefficients for the Cr$^{3+}$ atoms were taken from the Universal Force Field (UFF)[67]. The force field parameters for the bonded and Lennard-Jones terms involving the linker atoms were instead taken from the General Amber Force Field (GAFF)[68].

Water was modeled using the MB-pol many-body potential[33–35] that accurately reproduces the properties of water from gas-phase clusters to liquid water and ice[69,70]. As in our previous studies, the framework–water interactions were represented by electrostatic and LJ terms, except for the Cr$^{3+}$–water interactions where the LJ potential was replaced by a Buckingham potential fitted to ωB97X-D/def2-TZVP energy scans of a water molecule relative to the same cluster model used in the CM5 calculations (see Supplementary Fig. 3). The electrostatic term included both permanent and induced contributions where the water molecules were allowed to be polarized by the point charges of the framework. LJ parameters between the framework atoms and the water molecules were obtained according the Lorentz-Berthelot mixing rules using the LJ parameters of the TIP4P/2005 water model, which was shown to be the closest point-charge model to MB-pol[71].

**Molecular dynamics simulations.** All MD simulations were carried out under periodic boundary conditions for a system consisting of 2 × 2 × 2 primitive cells of NU-1500-Cr and various water loadings corresponding to RH values between 0% to 39.0%. MD simulations were performed in the isothermal-isobaric (constant number of atoms, pressure, and temperature, NPT); canonical (constant number of atoms, volume, and temperature, NVT); and microcanonical (constant number of atoms, volume, and total energy, NVE) ensembles to calculate various structural, thermodynamic, and dynamical properties. The temperature in the NVT and NPT simulations was controlled using a massive Nosé-Hoover chain thermostat where each degree of freedom was coupled to a Nosé-Hoover thermostat chain of length 4. The pressure in the NPT simulations was controlled by a Nosé-Hoover barostat based on the algorithm introduced in ref. [72]. The equations of motion were propagated according to the velocity-Verlet algorithm with a time step of 0.2 fs. The nonbonded interactions were truncated at an atom-atom distance of 9.0 Å, and the long-range electrostatic interactions were treated using the Ewald sum[73]. All MD simulations were carried out with in-house software based on the DL_POLY_2 simulation package. Assessments of the force field parameters are summarized in the Supplementary Tables 1–7.

At each loading, the initial configuration of the water molecules was prepared using Packmol, with a uniform distribution of the water molecules in all pores[74,75]. Each system was then further randomized in the NPT ensemble at 1 atm and 500 K for 100 ps, followed by 20 ps at 1 atm and 298.15 K. Lattice parameters, equilibrium bond distances, and enthalpies of adsorption were calculated from 1 ns trajectories carried out in the NPT ensemble at 1 atm and 298.15 K. Assuming that the water molecules in gas phase obey the ideal gas law, and the PV terms of MOF loaded with water and empty MOF are approximately equal, the enthalpy of adsorption at a given RH is given by

$$\Delta H_{ads} = \frac{U(\text{MOF} + \text{H}_2\text{O}) - U(\text{MOF}) - N \times U(\text{H}_2\text{O}) - N \times \text{RT}}{N}, \quad (2)$$

where N is the number of water molecules, R is the ideal gas constant, T is the temperature, U(MOF), U(H$_2$O), and U(MOF + H$_2$O) are the internal energies of empty MOF, single water molecule, and MOF loaded with water molecules at a given RH value, respectively. The dynamical properties and entropies were calculated by averaging over 20 independent 50 ps trajectories performed in the NVE ensemble, with the volume fixed at the average value calculated from the corresponding NPT simulations. The water entropy at each loading was calculated using the two-phase thermodynamic (2PT) model[76]. The theoretical infrared spectra were calculated from the autocorrelation function of the total dipole moment according to

$$I(\omega) = \left[\frac{2\omega}{3Vhce_0}\right] \tanh\left(\frac{\hbar\omega}{k_BT}\right) \int_{-\infty}^{\infty} \langle\mu(0)\mu(t)\rangle e^{i\omega t} dt, \quad (3)$$

where V is the volume of the system, c is the speed of light in the vacuum, $\epsilon_0$ is the vacuum permittivity, $k_B$ is the Boltzmann constant, T is the temperature, and $\langle\mu(0)\mu(t)\rangle$ is the ensemble averaged dipole-dipole time correlation function, with $\mu$ being represented by the many-body dipole moment function (MB-$\mu$)[77].

**Infrared spectroscopy.** NU-1500-Cr powder was mixed with KBr in a mass ratio of 1:20 and placed in a DiffusIR DRIFTS cell fitted to a Nicolet iS10 spectrometer. The humidity inside the DRIFTS cell was controlled by a humidity generator, which has previously been described[48], utilizing nitrogen gas and 18MΩ ultrapure water. The humidity inside the DRIFTS cell was monitored using a calibrated humidity meter, and each RH was maintained for 20 min before collecting a spectrum. Each measured FTIR spectrum was recorded immediately after collecting a background spectrum of KBr powder. A spectrum of activated NU-1500-Cr was obtained by activating the sample at 130 °C for several hours in a vacuum oven, transferring the sample to a nitrogen-filled DRIFTS cell while still above 120 °C, then acquiring the spectrum after the sample reached room temperature. Spectra of HOD in NU-1500-Cr were acquired by the same procedure as the H$_2$O spectra. Spectra were first acquired for a concentrated HOD solution made by combining ultrapure water and 99% D$_2$O (Cambridge Isotope Labs) in a 9:1 volume ratio. Spectra were then acquired for 99% D$_2$O without any added H$_2$O. The 99% D$_2$O spectra were used as backgrounds and subtracted from the concentrated HOD spectra to reduce the impact of C-H and other MOF peaks on the O-H band. The resulting

background-subtracted spectra are shown in Supplementary Figure 8 and were used for the fitting presented in Supplementary Fig. 9.

## Data availability

Any data generated and analyzed for this study that are not included in this Article and its Supplementary Information are available from the authors upon request.

## Code availability

The molecular models used in the MD simulations carried out with in-house software based on the DL_POLY_2 simulation package are publicly available on GitHub (https://github.com/paesanilab/Data_Repository/tree/main/NU-1500) in the format for the MBX[78] interface with LAMMPS[79]. All computer codes used in the analysis presented in this study are available from the authors upon request.

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

## Acknowledgements
We are grateful to Kelly Hunter, Hilliary Frank, Jierui Zhang, and Martina Lessio for stimulating discussions about computer simulations of MOFs, and Timur Islamoglu for discussions about the characterization of NU-1500-Cr. This research was supported by the Department of Energy, Basic Energy Science (BES) Office through awards no. DE-SC0022332 (C.-H.H., H.X., Z.C, O.K.F, and F.P.) and DE-SC0019333 (M.L.V. and W.X.). All simulations used resources of the National Energy Research Scientific Computing Center (NERSC), supported by Department of Energy BES Office under contract DE-AC02-05CH11231 and the Triton Shared Computing Cluster (TSCC) at the San Diego Supercomputer Center (SDSC).

## Author contributions
C.-H.H. performed simulations. M.L.V. performed infrared spectroscopy measurements. Z.C.and H.X. prepared and characterized the NU-1500-Cr sample. W.X. and F.P. conceived and designed research and administered the project. W.X., O.K.F, and F.P. acquired funding. C.-H.H., M.L.V., W.X., and F.P. analyzed and discussed the results and wrote the paper, with feedback from H.X., Z.C, and O.K.F.

## Competing interests
The authors declare no competing interests.
