## [Peer Review File · Communications Chemistry]

Reviewers' comments:

Reviewer #1 (Remarks to the Author):

Fantastic work! This manuscript by Paesani, Xiong, and Farha et al. presents the structure and thermodynamics of water molecules adsorbed in the channels of a MOF, NU-1500-Cr. The authors simulated the molecular dynamics (MD) of water molecules' hydrogen bond network residing in the NU-1500 MOF and then characterized the behavior of water molecules' dynamics and their network by using infrared spectroscopy. The strategy of this work was very well established and demonstrated, and thus, this paper is very well organized and well written. Therefore, I strongly support publishing this paper in this journal, Communications Chemistry. Before publishing this paper, however, I strongly recommend reading and referring to a recently published paper in this journal: Commun. Chem. 2022, 5, 51 (doi.org/10.1038/s42004-022-00666-8), which shows the hydrogen bonding network of water molecules, their chain connectivity, and ice-like behavior in the pores of HKUST-1.

Reviewer #2 (Remarks to the Author):

The work is of interest, although a bit narrow as it focuses very specifically on water adsorption in one MOF material, but it is a sound study with quite state-of-the-art methodology for the treatment of water-water interactions in a nanoporous framework. This will be a good paper, without a doubt, but at the moment the presentation is sometimes a mess, making the paper difficult to read. Moreover, some data is presented without being discussed or introduced, and it is sometimes unclear what is presented exactly.

- The authors present and discuss in Figure 1A an adsorption isotherm from the literature, although they have actually measured their own (Figure S11), but that is not discussed in the text. This is particularly surprising because the findings are quite different, with the authors' isotherm having a transition at RH between 33 to 40%, while that of Ref. 18 has a transition between 45 to 50%. What is the source of this difference? It should be plotted in the main text and discussed. It is particularly important because the authors' calculation clearly show no change in behavior in the 33 to 40% range, so there is an apparent discrepancy between the experimental isotherm and the calculations (unless I am missing something).

- Page 6: that the ΔH_{ads} is closer to the molar enthalpy of ice than liquid water does not necessarily imply anything about the strength of the H-bond network. The adsorption enthalpy also includes contributions from MOF-water interactions. If the authors want to compare the strength of the H-bond network, they need to compare the molar enthalpy of pure water to the sum of water-water interactions (not the adsorption enthalpy): it will be smaller, because the stabilization of the water inside the pores is the sum of water-water plus water-MOF interactions.

- Page 7: "Assuming that the entropy of the framework remains constant upon water adsorption" that seems like a very big ask. Adsorption of a dense phase is known to impact the entropy of the framework, and reduce its vibrational entropy because it restricts the molecular motions of the linkers. Given their approach, I think the authors could (if they want to back the claim made) compute the entropy of only the adsorbed phase?

- It is not clear to me whether the "bound" water molecules that are part of the SBU are treated in the simulations as adsorbed phase (meaning at RH = 0, the bound water molecules are removed) or part of the framework. Text on pages 7 and 8 clearly states that they are part of the framework, but the Methods section talk about the Cr³⁺ - water potential like those bound water molecules are part of

the adsorbate. Please clarify. Also, it seems to me that logic would dictate that all water molecules be treated the same... if it is not the case, the authors need to justify their choice.

- The study of the binding of water molecules with two specific populations arising from the nature of the open metal center reminds me of what is seen in MIL-53, whether experimentally (<https://doi.org/10.1021/jp202147m>) or through simulations (<https://doi.org/10.1039/C3CP53126K>), including some vibrational/spectroscopic characterization. It may be worth drawing a comparison with this system.

- I am a bit surprised by the mismatch between levels of theory used by the authors: the MBpol is clearly a very advanced level of description for the water, but the intramolecular MOF terms are of relatively low level of theory (GAFF, as any "universal" force field, performs quite poorly on individual materials when you look at detailed properties), and the water-MOF interactions are treated with generic Lorentz-Berthelot mixing rules. I think it would be interesting (not only for me, but for all readers) for the authors to comment on this.

- From the methods (page 22), it is not clear how the electrostatic terms for water-MOF interactions are determined. The text says "The electrostatic term included both permanent and induced contributions where the water molecules were allowed to be polarized by the point charges of the framework." How are the charges for framework atoms determined?

- "The molecular models used in the MD simulations carried out with in-house software based on the DL POLY 2 simulation package are available in the format for the MBX77 interface with LAMMPS78 from the authors upon request." -- given the complexity of the force fields involved, it is necessary in my view for the authors to publish the input files (force fields, control files, initial configurations) for at least each type of calculation performed. Unless the input files are public there is no way for other groups to reproduce the work reported here. (The computer code is already not public, which I guess is something the authors do not want to change, sadly.)

- page 23, "Each system was then further randomized in the NPT ensemble": what does "randomized" in this context mean? Equilibrated?

- page 23, before equation 2: "the PV terms of MOF loaded with water and empty MOF are approximately equal". Wouldn't the compressibility of the empty and loaded MOF be different? Can the authors estimate how good that approximation is? I think it might be true because the pressures here are really small (for a solid), but it would be good to have an estimation.

- The number of supplementary figures in the text is sometimes inconsistent, shifted from the actual numbers (for example Figures S9 and S10 in the main text are actually S10 and S11).

Response to Reviewer 1

Fantastic work! This manuscript by Paesani, Xiong, and Farha et al. presents the structure and thermodynamics of water molecules adsorbed in the channels of a MOF, NU-1500-Cr. The authors simulated the molecular dynamics (MD) of water molecules' hydrogen bond network residing in the NU-1500 MOF and then characterized the behavior of water molecules' dynamics and their network by using infrared spectroscopy. The strategy of this work was very well established and demonstrated, and thus, this paper is very well organized and well written. Therefore, I strongly support publishing this paper in this journal, Communications Chemistry. Before publishing this paper, however, I strongly recommend reading and referring to a recently published paper in this journal: Commun. Chem. 2022, 5, 51 (doi.org/10.1038/s42004-022-00666-8), which shows the hydrogen bonding network of water molecules, their chain connectivity, and ice-like behavior in the pores of HKUST-1.

We thank the Reviewer for the positive assessment of our manuscript, and support for publication.

We added the reference to the paragraph that discusses the results presented in Figure 3(A). The revised sentence reads as: "A similar conclusions were drawn from computer simulations of water in MIL-100(Fe), MIL-101(Cr), and Co₂Cl₂BTDD, and a recent experimental study of water in HKUST-1."

Response to Reviewer 2

The work is of interest, although a bit narrow as it focuses very specifically on water adsorption in one MOF material, but it is a sound study with quite state-of-the-art methodology for the treatment of water-water interactions in a nanoporous framework. This will be a good paper, without a doubt, but at the moment the presentation is sometimes a mess, making the paper difficult to read. Moreover, some data is presented without being discussed or introduced, and it is sometimes unclear what is presented exactly.

We thank the Reviewer for a careful reading of our manuscript and their comments/suggestions, which provide us with the opportunity to clarify some of our discussions and improve the presentation of our results.

The authors present and discuss in Figure 1A an adsorption isotherm from the literature, although they have actually measured their own (Figure S11), but that is not discussed in the text. This is particularly surprising because the findings are quite different, with the authors' isotherm having a transition at RH between 33 to 40%, while that of Ref. 18 has a transition between 45 to 50%. What is the source of this difference? It should be plotted in the main text and discussed. It is particularly important because the authors' calculation clearly show no change in behavior in the 33 to 40% range, so there is an apparent discrepancy between the experimental isotherm and the calculations (unless I am missing something).

We thank the Reviewer for raising this point, which gave us the opportunity to review the original measurements. The experimental measurements presented in this study were taken by the same group that reported the original isotherm of adsorption in J. Am. Chem. Soc. 141, 2900–2905 (2019). It turned out that in original isotherms P_0 was (mistakenly) set to 2340 Pa, which is the value of P_0 at 20 °C, instead of 3169 Pa, which is the value of P_0 at 25 °C. As a consequence, the value of P/P_0 corresponding to water condensation reported in the original study is incorrect, which is responsible for the shift in the adsorption step noted by the Reviewer. This also explains the apparent inconsistency between the simulation results and the isotherm of adsorption reported in our study because the number of water molecules used in the MD simulations was calculated based on the original isotherm of adsorption. It is important to note that, since the number of water molecules used in the simulations was determined from the weight percentage, the different values of P_0 used in the two experiments do not affect the reliability of our MD simulations but only "rigidly" shifts the RH axis.

In order to maintain consistency between the experimental and simulation data presented in our study, we replaced the original isotherm of adsorption shown in Figure 1(A) with the new isotherm shown in Figure S11, and revised the text throughout the manuscript according to the correct values of RH.

Page 6: that the ΔH_{ads} is closer to the molar enthalpy of ice than liquid water does not necessarily imply anything about the strength of the H-bond network. The adsorption enthalpy also includes contributions from MOF-water interactions. If the authors want to compare the strength of the H-bond network, they need to compare the molar enthalpy of pure water to the sum of water-water interactions (not the adsorption enthalpy): it will be smaller, because the stabilization of the water inside the pores is the sum of water-water plus water-MOF interactions.

We completely agree with the Reviewer. In fact, the analysis of the q_{tet} order parameter shows that water adsorbed in the NU-1500 pores does not properly resemble ice. Our goal indeed was to make a contraposition

between the enthalpy values and hydrogen-bonding local motifs but, based on the Reviewer’s comment, we realized that our original statement may cause some confusion. We, therefore, decided to replace the original sentence as follows: “It is worth noting that ΔH_{ads} at high RH values is more negative than the enthalpy calculated from MB-pol simulations of liquid water at 298 K (~ -10.96 kcal/mol), which suggests that the confinement of water molecules within the NU-1500-Cr framework is, overall, energetically favorable.”

Page 7: "Assuming that the entropy of the framework remains constant upon water adsorption" that seems like a very big ask. Adsorption of a dense phase is known to impact the entropy of the framework, and reduce its vibrational entropy because it restricts the molecular motions of the linkers. Given their approach, I think the authors could (if they want to back the claim made) compute the entropy of only the adsorbed phase?

We have realized that the language used in the original manuscript may be confusing. By construction, the 2PT model used in our study is only applicable to the adsorbed phase (in our case water) and, therefore, our calculations exactly report what the Reviewer is asking us to calculate.

Regarding the sentence that the Reviewer refers to, the first part of the sentence was meant to point out that the variation of the water entropy calculated with the 2PT method closely follows the adsorption isotherm, which suggests that the entropy of the framework remains approximately constant throughout the adsorption process. However, since we directly calculated the entropy of water without making any assumption for the entropy of the framework, we removed the first part of the original sentence in order to avoid any possible confusion.

It is not clear to me whether the “bound” water molecules that are part of the SBU are treated in the simulations as adsorbed phase (meaning at RH = 0, the bound water molecules are removed) or part of the framework. Text on pages 7 and 8 clearly states that they are part of the framework, but the Methods section talk about the Cr^{3+} - water potential like those bound water molecules are part of the adsorbate. Please clarify. Also, it seems to me that logic would dictate that all water molecules be treated the same... if it is not the case, the authors need to justify their choice.

All water molecules in our simulations are equivalent and “feel” the same interactions with the framework. At the beginning of the simulations, there is no distinction between “bound” and “unbound” water molecules. A water molecule becomes “bound” when it binds to one of the Cr^{3+} sites. As described in the Methods section, the potential energy function describing the Cr^{3+} -water interaction was specifically parameterized to reproduce $\omega\text{B97X-D/def2-TZVP}$ reference energies calculated for radial scans of a water molecule relative to the Cr^{3+} site using a cluster model of the SBU (Figure S1).

The original Figure S3 contained a correlation plot demonstrating the accuracy of our force field in replicating the Cr^{3+} -water interaction. However, we have updated Figure S3 to not only showcase the accuracy of our fitting, but also to depict the water molecule orientations in relation to the SBU.

The study of the binding of water molecules with two specific populations arising from the nature of the open metal center reminds me of what is seen in MIL-53, whether experimentally (<https://doi.org/10.1021/jp202147m>) or through simulations (<https://doi.org/10.1039/C3CP53126K>), including some vibrational/spectroscopic characterization. It may be worth drawing a comparison with this system.

We appreciate the suggestion regarding the comparison with MIL-53 for broader insights. However, MIL-53(Cr) and NU-1500-Cr have significantly different structural properties, even though they share some similarity in the properties of the Cr sites. For example, MIL-53(Cr) undergoes breathing as a function of water loading, which makes it difficult to compare with NU-1500-Cr whose structure remains effectively unchanged. As a result, we referenced some other frameworks that do not undergo structural changes, such as MIL-100(Fe), MIL-101(Cr), and $\text{Co}_2\text{Cl}_2\text{BTDD}$ in the text, to provide additional insights.

In this regard, we also note that we use our many-body models to characterize both linear and nonlinear infrared spectra of water in MIL-53(Cr) [J. Phys. Chem. Lett. **5**, 2897 (2014)], which clearly show a different evolution as a function of RH from the infrared spectra measured and simulated in our study for water in NU-1500(Cr). As mentioned above, these differences can be traced back to both differences in the structural properties of the two frameworks (e.g., shape, size, and flexibility) and differences in the nature of framework-water interactions (e.g., μ_{OH} -water vs. Cr^{3+} -water interactions).

I am a bit surprised by the mismatch between levels of theory used by the authors: the MBpol is clearly a very advanced level of description for the water, but the intramolecular MOF terms are of relatively low level of theory (GAFF, as any "universal" force field, performs quite poorly on individual materials when you look at detailed properties), and the water-MOF interactions are treated with generic Lorentz-Berthelot mixing rules. I think it would be interesting (not only for me, but for all readers) for the authors to comment on this.

We thank the Reviewer for this comment that allows us to clarify some possible confusion. Although, as recognized by the Reviewer, the level of accuracy and sophistication of our data-driven many-body MB-pol model is somewhat unique, framework-water interactions in our simulations are also described according to a more advanced and accurate scheme than that adopted by all simulations of water in MOFs reported in

the literature which rely on popular pairwise-additive force fields. In particular, our simulations are fully many-body in nature since, in addition to polarize each other, the MB-pol water molecules are also polarized by the framework. This implies that water molecules located in different regions within the framework acquire a different induced dipole moment, which is key to correctly represent specific framework–water interactions. This is completely different from all simulations of water in MOFs reported in the literature using pairwise-additive force fields in which the dipole moment of a water molecule is fixed, independently of the location of the water molecule within the MOF and RH values.

In addition, as discussed above and in the Methods section of our manuscript, the Cr^{3+} –water interaction, which is the key interaction that determines the initial steps of water adsorption in NU-1500, was directly fitted to *ab initio* energies. It thus follows that in our simulations the water–MOF interactions are not simply treated with generic Lorentz-Berthelot mixing rules, which are instead only used on top of an accurate representation of many-body interactions to describe small nonbonded contributions associated with linker–water interactions. The accuracy of our representation of the framework–water interactions is further demonstrated by the close agreement between the experimental and simulated infrared spectra of water as a function of RH.

In the context of computer simulations of water in MOFs, we would also like to emphasize that some caution should also be used when performing MD simulations of water in MOFs within density functional theory (DFT) as those reported in <https://doi.org/10.1039/C3CP53126K> mentioned by the Reviewer. It has now been established that common density functionals developed within the generalized gradient approximation (i.e., GGA functionals) as the PBE functional used in <https://doi.org/10.1039/C3CP53126K> suffers from intrinsic functional-driven and density-driven errors [e.g., see *Phys. Rev. Lett.* **100**, 146401 (2008); *Phys. Rev. Lett.* **111**, 073003 (2013); *J. Am. Chem. Soc.* **144**, 6625 (2022)], which drastically limit their accuracy and transferability when applied to water [e.g., *Proc. Natl. Acad. Sci. U.S.A.* **117**, 11283 (2020); *Nat. Commun.* **12**, 6359 (2021); *J. Chem. Theory Comput.* **18**, 3410 (2022)]. In this context, MD simulations carried out with MB-pol currently provide the most realistic representation of water across different phases and thermodynamic state points [e.g., see <https://doi.org/10.26434/chemrxiv-2023-kmmmz>].

From the methods (page 22), it is not clear how the electrostatic terms for water-MOF interactions are determined. The text says "The electrostatic term included both permanent and induced contributions where the water molecules were allowed to be polarized by the point charges of the framework." How are the charges for framework atoms determined?

As discussed in the Methods section, the charges of atoms in the framework were obtained using the Charge Model 5 as implemented in Gaussian 16 by performing DFT calculations with on cluster models shown in Supplementary Figures 1 and 2. As mentioned above, being polarizable by construction, MB-pol water molecules in our simulations are able to respond to variations in the electric field generated by the framework and other water molecules.

"The molecular models used in the MD simulations carried out with in-house software based on the DL POLY 2 simulation package are available in the format for the MBX77 interface with LAMMPS78 from the authors upon request." – given the complexity of the force fields involved, it is necessary in my view for the authors to publish the input files (force fields, control files, initial configurations) for at least each type of calculation performed. Unless the input files are public there is no way for other groups to reproduce the work reported here. (The computer code is already not public, which I guess is something the authors do not want to change, sadly.)

We are a bit confused by this Reviewer’s comment. As stated in data availability and code availability sections, all data generated and analyzed for this study are available upon request. Moreover, while our simulations were carried out with an in-house version of DL_POLY2 because this project began before we completed the interface of our open-source MBX software (<https://github.com/paesanilab/MBX>) with LAMMPS, the same simulations can now be carried out with MBX+LAMMPS as stated in our manuscript. For this reason, we made our molecular models available on GitHub in the format used by MBX+LAMMPS so that everyone can directly use them in their MD simulations and replicate all results reported in our study.

Page 23, "Each system was then further randomized in the NPT ensemble": what does "randomized" in this context mean? Equilibrated?

In order to generate different initial configurations for the water molecules in the NU-1500 pores at different temperatures and RH values, as explained in the method section, we performed short MD simulations in the NPT ensemble at high temperature (500 K), starting from the Packmol configurations. These NPT simulations effectively “randomize” the positions of the water molecules in the pores.

Page 23, before equation 2: "the PV terms of MOF loaded with water and empty MOF are approximately equal". Wouldn’t the compressibility of the empty and loaded MOF be different? Can the authors estimate how good that approximation is? I think it might be true because the pressures here are really small (for a solid), but it would be good to have an estimation.

The PV difference between empty and filled NU-1500 is ~ 0.06 kcal/mol at 1 atm over the RH values analyzed in our study.

The number of supplementary figures in the text is sometimes inconsistent, shifted from the actual numbers (for example Figures S9 and S10 in the main text are actually S10 and S11).

We thank the Reviewer for catching this. The figure numbers have been corrected accordingly in the revised manuscript.

REVIEWERS' COMMENTS:

Reviewer #2 (Remarks to the Author):

The revisions made were minimal, and there is a lot of discussion given in the rebuttal letter that has not been incorporated into the manuscript itself (or SI). I think most of the points raised are of value, and should at least be clarified in the text, and discussed where relevant, because other readers might have the same questions as I had, and would benefit from this.

Regarding the comparison with previous experimental data, even if those were performed by the same group I do not think it is acceptable to publish contradicting data with actually drawing a comparison: the explanation for the original error is clear, and should be stated clearly in this paper when the data is presented (unless a correction has already been published for the original paper).

Response to Reviewer 2

The revisions made were minimal, and there is a lot of discussion given in the rebuttal letter that has not been incorporated into the manuscript itself (or SI). I think most of the points raised are of value, and should at least be clarified in the text, and discussed where relevant, because other readers might have the same questions as I had, and would benefit from this.

Regarding the comparison with previous experimental data, even if those were performed by the same group I do not think it is acceptable to publish contradicting data with actually drawing a comparison: the explanation for the original error is clear, and should be stated clearly in this paper when the data is presented (unless a correction has already been published for the original paper).

We thank the Reviewer for these additional comments.

We have added an explanation of the difference between the original isotherm of adsorption reported in *J. Am. Chem. Soc.* 141, 2900 (2019) and the isotherm of adsorption measured in this study at the beginning of the beginning of the section “Thermodynamics of water adsorption”.

Regarding the other comments, we believe that the Methods section of our manuscript already includes all the details about both the models and the computational methods used in the simulations. In particular, we would like to emphasize that MB-pol is a well established water model, which has been discussed in detail in the literature [e.g., *J. Chem. Theory Comput.* 9, 12, 5395 (2013); *J. Chem. Theory Comput.* 10, 1599 (2014); *J. Chem. Theory Comput.* 10, 2906 (2014)] and used in simulations of various systems, including MOFs [e.g., *Nat. Commun.* 10, 4771 (2019); *J. Phys. Chem. C* 125, 12451 (2021); *J. Am. Chem. Soc.* 143, 21189 (2021)]. All previous applications of MB-pol relevant to our study are included in our manuscript.